# Reproducibility of predictive networks for mouse visual cortex

**Polina Turishcheva,**[1,†] **Max F. Burg,**[1-3] **Fabian H. Sinz,**[1-2] **Alexander S. Ecker**[1,4,*]

[1] Institute of Computer Science and Campus Institute Data Science, University Göttingen, Germany
[2] International Max Planck Research School for Intelligent Systems, Tübingen, Germany
[3] Tübingen AI Center, University of Tübingen, Germany
[4] Max Planck Institute for Dynamics and Self-Organization, Göttingen, Germany

[†]`turishcheva@cs.uni-goettingen.de`, [*]`ecker@cs.uni-goettingen.de`

## Abstract

Deep predictive models of neuronal activity have recently enabled several new discoveries about the selectivity and invariance of neurons in the visual cortex. These models learn a shared set of nonlinear basis functions, which are linearly combined via a learned weight vector to represent a neuron's function. Such weight vectors, which can be thought as embeddings of neuronal function, have been proposed to define functional cell types via unsupervised clustering. However, as deep models are usually highly overparameterized, the learning problem is unlikely to have a unique solution, which raises the question if such embeddings can be used in a meaningful way for downstream analysis. In this paper, we investigate how stable neuronal embeddings are with respect to changes in model architecture and initialization. We find that $L_1$ regularization to be an important ingredient for structured embeddings and develop an adaptive regularization that adjusts the strength of regularization per neuron. This regularization improves both predictive performance and how consistently neuronal embeddings cluster across model fits compared to uniform regularization. To overcome overparametrization, we propose an iterative feature pruning strategy which reduces the dimensionality of performance-optimized models by half without loss of performance and improves the consistency of neuronal embeddings with respect to clustering neurons. Our results suggest that to achieve an objective taxonomy of cell types or a compact representation of the functional landscape, we need novel architectures or learning techniques that improve identifiability. The code is available `https://github.com/pollytur/readout_reproducibility`.

## 1 Introduction

One central idea in neuroscience is that neurons cluster into distinct cell types defined by anatomical, genetic, electrophysiological or functional properties of single cells [23, 49, 54, 59, 69, 70]. Defining a unified taxonomy of cell types across these different descriptive properties is an active area of research. The functional properties of neurons, i.e. which computations they implement on the sensory input, constitute an important dimension along which cell types should manifest. However, each neuron's function is a complex object, mapping from a high-dimensional sensory input to the output of the neuron. Classical work in neuroscience has described neuronal function by a few parameters such as tuning to orientation, spatial frequency, or phase invariance [1, 2, 5, 8, 9, 13, 25, 40]. However, when considering neurons from higher visual areas, deeper into the brain's processing network, neuronal functions become more complex and not easily described by few

38th Conference on Neural Information Processing Systems (NeurIPS 2024).

manually picked parameters. In addition, recent work in the mouse visual cortex demonstrated that even in early areas, neuronal functional properties are not necessarily well-described by simple properties such as orientation [47, 71]. One possible avenue to address these challenges is to use functional representations learned by data-driven deep networks [36, 37, 39, 44] trained to predict neuronal responses from sensory inputs. The central design element of these networks is a split into a *core* – a common feature representation shared across neurons, and a neuron specific *readout* – typically a linear layer with a final nonlinearity (see Fig. 1). These networks are state-of-the-art in predicting neuronal responses on arbitrary visual input, and several studies have used these networks as "digital twins" to identify novel functional properties and experimentally verify them *in vivo* [37, 47, 57, 58, 67, 68]. Because readout weights determine a compact representation of a neuron's input-output mapping, they can serve as an embedding of a neuron's function. Such embeddings could be used to describe the functional landscape of neurons or obtain cell types via unsupervised clustering [59]. However, because the feature representations provided by the core are likely overcomplete, there will be many readout vectors that represent approximately the same neuronal function. Thus, it is not clear how identifiable functional cell types are based on clustering these embeddings. Moreover, since early data-driven networks [22, 28] there have been several advances in readout [48, 51, 65] and network architecture [33, 36, 56, 67] to decrease the number of per-neuron parameters, incorporate additional signals such as behavior and eye movements, or add specific inductive biases, such as rotation equivariance. Currently, there is no study systematically investigating the impact of these architectural choices on the robustness and the consistency of embedding-based clustering.

Here we address this question by quantifying how consistent the models are across several fits with different seeds. We measure consistency by (1) adjusted rand index (ARI) of clustering partitions across models, (2) correlations of predicted responses across models, and (3) consistency of tuning indexes describing known nonlinear functional properties of visual neurons. Our contributions are:

- We show that $L_1$ regularization, used in early models [28, 59], is instrumental in obtaining a structured clustering when using embeddings of newer readout mechanisms [51].

- We introduce a new adaptive regularization scheme, which improves consistency of learned neuron embeddings while maintaining close to the state-of-the-art predictive performance.

- We address the identifiability problem by proposing an iterative feature pruning strategy which reduces the dimensionality of performance-optimized models by half without loss of performance and improves the consistency of neuronal embeddings.

- We show that even though our innovations improve the consistency of clustering neurons, older readout mechanisms [28] reproduce neuronal tuning properties more consistently across models.

Our results suggest that to achieve an objective taxonomy of cell types or a compact representation of the functional landscape, we need novel architectures or learning techniques that improve identifiability while preserving neuronal functional properties.

## 2 Background and related work

**Predictive models for visual cortex.** Starting with the work of Antolík et al. [22], a number of deep predictive models for neuronal population responses to natural images have been proposed. These models capture the stimulus-response function of a population of neurons by learning a nonlinear feature space shared by all neurons (Fig. 1A; the "core"), typically implemented by a convolutional neural network [26, 28, 51], sometimes including recurrence [36, 55]. The core can be pretrained and shared across datasets [67]; it forms a set of basis functions spanning the space of all neuron's input-output function. The second part of these model architectures is the "readout": it linearly combines the core's features using a set of neuron-specific weights.

As the dimensionality of the core's features is quite high (height $\times$ width $\times$ feature channels), several different ways have been proposed to constrain and regularize the readout, accounting to the fact that neurons do not see the whole stimuli but rather focus on a small receptive field (RF). Initially, Klindt et al. [28] proposed a factorization across space (RF location) and features (function) (Fig. 1A; "Factorized readout"). This approach required relatively strong $L_1$ regularization to constrain the RF. More recently, Lurz et al. [51] further simplified the problem by learning only an $(x, y)$ coordinate, representing the mean of a Gaussian for the receptive field location, as well as a vector of feature

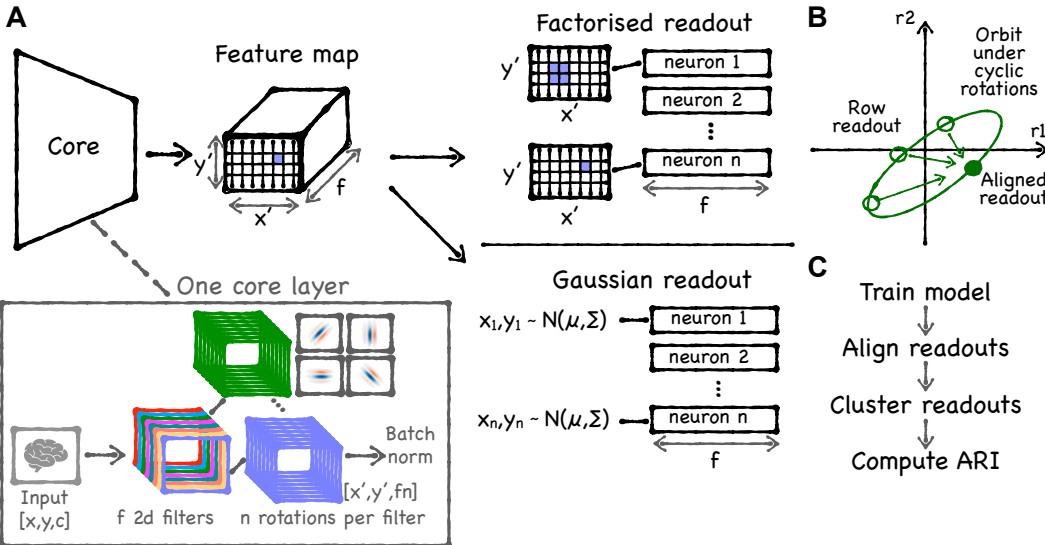

Figure 1: **A. Model architecture**: The models used in our study are separated into a core learning a non-linear feature representation shared across neurons, and a neuron specific linear readout. In our case, the core is a rotation-equivariant core, i.e. each learnt feature $f$ is analytically rotated $n$ times by $360/n$ degrees, resulting in $f \cdot n$ output channels. We apply batch norm on the learnt feature channels $f$ only and do not learn scale and bias parameters in the core's last layer, as this would interfere with the readout regularization. Two commonly used linear readouts are factorized and Gaussian readout. The factorized readout learns spatial weights accounting for neuron's receptive field position and feature weightings – the neural embeddings. The Gaussian readout replaced the spatial weights by sampling receptive field positions from a Gaussian and at inference time samples the Gaussian's mean. **B. Readout alignment**: To yield rotation invariant neural embedding clusters, we align the orientations of the embedding vectors by cyclically rotating the elements in the embedding vectors to minimize the sum of pairwise distances between the rotated embeddings [46].**C. Our pipeline**: For each model architecture and training configuration we train 3 models with different parameter initialization seeds. We orientation-align their embeddings, cluster them, and eventually compute all model pair-wise ARIs.

weights (Fig. 1A; "Gaussian readout"). This removed the necessity for strong $L_1$ regularization. Current state-of-the-art models use Gaussian readout with weak or no regularization [60, 64, 67].

**Functional properties.**   Previous work verified these model in vivo [37, 47, 58, 67, 68]. There is a long history of work investigating neuronal functional properties experimentally, beginning with Hubel and Wiesel (1962) [1] who showed bar stimuli to cats and found that many neurons in V1 are orientation selective. Since then, a number of studies showed that many V1 neurons are – in a linear model approximation – optimally driven by a Gabor stimulus of specific orientation, size, spatial frequency, and phase [6, 20, 32, 40]. Later, non-linear phenomena were investigated, such as surround suppression and cross-orientation inhibition. Neurons that are surround suppressed can be driven by stimulating their receptive field, while an additional stimulation of their surround reduces neural activity. Interestingly, only stimulating the surround would not elicit a response. Cross-orientation inhibition reduces neural activity of some neurons if a neuron's optimal Gabor stimulus is linearly combined with a 90 degrees rotated version of that Gabor stimulus, forming a plaid [3, 5, 7, 8, 10, 14, 17]. Recently, such experiments were performed in silico with neural predictive models, allowing for large-scale analysis of neuron's tuning properties without experimental limitations [53, 59].

**Network pruning.**   While modern deep neural networks are typically highly overparameterized, as this simplifies optimization [38, 50, 52], overparametrization poses a problem for reproducibility of neural embeddings: model training might end in various local optima that leads to different neuronal representations even if model outputs and predictive performance are similar. Prior work has shown that learned deep neural networks can be pruned substantially: Frankle and Carbin [34] showed

that up to 80% of weights can be removed without sacrificing model performance. Further work investigated various pruning strategies. For instance Li et al. [29] pruned CNN channels based on their $L_1$ norm, Luo and Wu [30] used entropy as the pruning criteria, and various other criteria were studied [24, 27, 31]. Interestingly, a later study showed that model performance is relatively robust against the specific pruning strategy used [35].

## 3 Data and model architecture

**Data.** We used the data from the NeurIPS 2022 Sensorium Competition for our study [60]. This dataset contains responses of primary visual cortex (V1) neurons to gray-scale natural images of seven mice, recorded using two-photon calcium imaging. In addition the mice's running speed, pupil center positions, pupil dilation and its time-derivative were recorded.

**Model architecture.** Our model (Fig. 1) builds upon the baseline model from the NeurIPS 2022 Sensorium competition [60]. Similar as the baseline model, our model includes the *shifter* sub-network [36] to account for eye movements and correct the resulting receptive field displacement. We also concatenate the remaining behavioral variables to the stimuli as the input to the model [56, 60]. For the model core, we use a rotation-equivariant convolutional neural network, roughly following earlier work [33, 46, 59]. Compared to this prior line of work, our model includes an additional convolutional layer (as the dataset is larger). It consists of four layers with filter sizes 13, 5, 5, and 5 pixels. In all layers, we use sixteen channels and eight rotations for each of the channels, which results in 128-dimensional neuronal embeddings. The details of model and training config are provided in Appendix A.1. For the readout, we use both the factorized readout and the Gaussian readout (Fig. 1A). The factorized readout has been used in prior work on functional cell types [46, 59]. As it does not support the shifter network, this version of the model cannot account for eye movements.

## 4 Methods

**Training.** Following prior work [28, 33, 36, 47, 53, 58, 60, 64, 66], we trained the model minimizing the Poisson loss $L_p = N^{-1} \sum_{i=1}^{N} (\hat{r}^{(i)} - r^{(i)} \log \hat{r}^{(i)})$ between predicted $\hat{r}^{(i)}$ and observed $r^{(i)}$ spike counts for all $i = 1, ..., N$ neurons, as it is oftentimes assumed that neuron's firing rates follow a Poisson process [15]. For factorized readouts we add a $L_1$ regularization of the readout mask and embeddings to the loss, i.e. $L_1 = \sum_{i=0}^{N} \gamma \left( \sum_{i,j}^{\text{mask}} |m_{ij}| + \sum_{k}^{\text{embedding}} |w_k| \right)$ for spatial mask $m$, embedding $w$, and regularization coefficient $\gamma$, in line with previous work [28, 33, 39, 46, 53, 59]. As the Gaussian readout selects one spatial position, the $L_1$ penalty term reduces to $L_1 = \sum_{i=0}^{N} \gamma \sum_{k}^{\text{embedding}} |w_k|$. Thus, for both models the total loss becomes $Loss = L_p + L_1$. For the *factorized* readout $\gamma$ is typically chosen to maximize performance (Fig. 6), while the choice of $\gamma$ for *Gaussian* readouts is the subject of this paper.

**Evaluation of model performance.** Following earlier work [21, 33, 36, 47, 53, 60, 64, 66], we report model's predictive performance using Pearson correlation between each measured and predicted neural activity pair across images on the test set.

**Evaluation of embedding consistency.** Assessing how "consistent" neuronal embeddings are across model architectures and initializations is a non-trivial question. They will naturally differ across runs in absolute terms, because core and readout are learned jointly. In the context of identifying cell types, we are interested in the relative organization of the embedding space: Will the same sets of neurons consistently cluster in embedding space? To address this question, we use clustering and evaluate how frequently pairs of neurons end up in the same group by computing the adjusted rand index (ARI, [4]). The ARI quantifies the similarity of two cluster assignments $X$ and $Y$:

$$ARI = \frac{\sum_{ij} \binom{n_{ij}}{2} - \left[ \sum_i \binom{a_i}{2} \sum_j \binom{b_j}{2} \right] \Big/ \binom{n}{2}}{\frac{1}{2} \left[ \sum_i \binom{a_i}{2} + \sum_j \binom{b_j}{2} \right] - \left[ \sum_i \binom{a_i}{2} \sum_j \binom{b_j}{2} \right] \Big/ \binom{n}{2}} . \tag{1}$$

Here, $a_i$ is the number of data points in cluster $i$ of partition $X$, $b_j$ is the number of data points in cluster $j$ of partition $Y$, $n_{ij}$ is the number of data points in clusters $i$ and $j$ of partition $X$ and $Y$,

respectively, and $n$ is the total number of data points. The ARI is invariant under permutations of the cluster labels. It is one if and only if the two partitions match exactly. It is zero when the agreement between the two partitions is as good as expected by chance and, hence, can be negative [61].

For clustering, we follow the rotation-invariant clustering procedure (Fig. 1B) introduced by Ustyuzhaninov et al. [46] and then cluster the aligned embeddings with k means. As we do not know the true underlying number of clusters, we explore a range of $k = 5$ up to $k = 100$ clusters, which covers the typical amount of cell types suggested in various works and modalities [41, 42, 45, 59].

To visualize the neuronal embeddings, we use t-SNE [16] using the recommendations of Kobak and Berens [43]. We use a perplexity given by 1% of the size of the dataset and set early exaggeration to 15. To avoid visual clutter, we plot a randomly sampled subset of 2,000 neurons from each of the seven animals in the dataset. We use the same neurons across all ARI experiments and t-SNE visualizations. The results do not depend on the specific cluster consistency metric used (see Appendix A.9). We assessed the credibility of t-SNE visualizations in Appendix A.10.

**Neuronal tuning properties.**   Unlike the embeddings, which were not yet biologically validated, the predicted activity from our models has been consistently verified in vivo across several studies [47, 58, 59, 68]. By focusing on predicted activity, we leverage the stability and biological relevance of these outputs and relate our findings to real-world scenarios. Specifically, we investigated the predicted neural activity to parametric, artificial stimuli that are known to yield interesting phenomena and were used extensively in biological experiments (see Section 2).

To quantify neuron's tuning properties, for *phase or orientation tuning*, we use the parameters of the optimal Gabor and change only the phase or orientation to create a set of new Gabors. We feed this set of stimuli through the model and fit sinusoidal functions to the stimulated response curve, namely $f(\alpha) = A \sin(\alpha + \phi) + B$ for the phase sweep. For the orientation sweep, rotating the Gabor-based stimuli by 180-degrees will lead the same stimulus, accounted for by the additional factor of two when fitting $f(\alpha) = A \sin(2\alpha + \phi) + B$ to the stimulated responses. We define the corresponding tuning indices as the ratio of the fitted amplitude $A$ to the mean $B$ of the curve.

For *cross-orientation inhibition* we added an optimal Gabor and its orthogonal copy and stimulated the tuning curve by varying the contrast of the orthogonal Gabor. Analogically, for *surround suppression* we varied the size and the contrast of a grating based on the optimal Gabor's parameters. To quantify both indices, we followed [11, 53] and saved the highest model prediction $\hat{r}_{max}$. Next, we computed the according tuning indices as $1 - \hat{r}_{supp}/\hat{r}_{max}$, where $\hat{r}_{supp}$ corresponds to the highest suppression/inhibition observed. See Appendix Section A.2 for parameter details.

We compute tuning index consistency across three trained models only differing in their parameter initialisation before fitting. Specifically, we compare their similarity by the normalized mean absolute error $NMAE(x, y) = \sum_i^{neurons}((|x_i - y_i|)/(|x_i - \bar{x}|))$ where $x$ and $y$ are tuning indices for different model fits and $\bar{x}$ denotes the mean across neurons.

**Optimal Gabor search.**   All tuning properties investigated in our paper are based on optimal Gabor parameters. Previous work [33, 53, 59] brute-force searched the optimal stimulus. Since their models relied on the factorized readout which could integrate over multiple spatial input positions, they had to show Gabor stimuli of various properties at all spatial input positions. Thus, they needed to iterate through millions of artificially generated Gabor stimuli for each neuron. Here, we introduce a technical improvement upon their work significantly speeding up the optimal Gabor search for Gaussian readouts: as this readout type selects only one spatial position per neuron it is sufficient to limit the presented stimulus canvas size to the model's receptive field size. This significantly decreased the canvas size from $36 \times 64 = 2,304$ to $25 \times 25 = 625$ input pixels, leading to an approximately 3.7-fold speedup. Additionally, our improved approach reduces the number of floating point operations in the model's forward pass, further benefiting the computation time. Another important detail is that for the optimal Gabor search and tuning curves experiments we completely ignored the learnt neuronal position as there was only one position to choose. For all in silico experiments, we removed models' *shifter* pupil position prediction network [36], as pupil position only impacted the spacial position selection, which is now limited to a single pixel. All other behavior variables we set to their training set median values, following [56, 67].

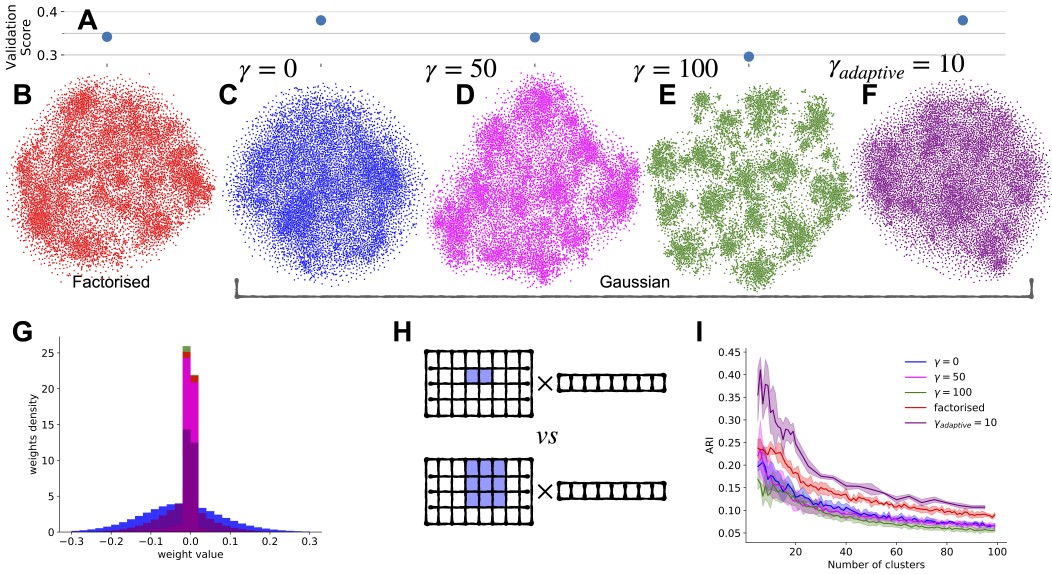

Figure 2: **A:** Model performance: correlation between predicted and observed neuronal response. **B–F:** T-SNE projections of the embeddings from differently regularized models. All models used $seed = 42$. **G:** The histogram of weights before alignment from different models. Unregularized weights ($\gamma = 0$) are crucially different, while $\gamma = 50$ and $\gamma = 100$ are close to the factorized readout's distribution. $\gamma_{lognorm} = 10$ is not as sparse as the factorized one. **H:** Example of 'adaptive' regularization for the factorized mask. The "mask" is a 2D matrix, selecting a "receptive field" from the latent space and the "embedding" is a linear vector representing a learned neuron function. Both are weights learned in the readout. In the top, the mask learnt is much smaller, hence, regularizing a neuronal embedding is more important to reduce the $L_1$ penalty, while in the bottom we can keep the neuronal embedding less sparse; its impact is smaller as the mask is bigger. **I:** Adjusted rand index (ARI) for clustering embeddings using $k$-means. We take embeddings from models trained with different seeds, cluster them and compute ARI between the clusterings. Note that even clustering the same embeddings twice with different clustering initializations will result in ARI $< 1$ (see Appendix A.4).

## 5   Results

**From factorized to Gaussian readouts.**   We start with our observation that the structure of the neuronal embeddings depends quite strongly on the type of readout mechanism employed by the model (Fig. 2). Earlier studies employing the factorized readout reported clusters in the neuronal embeddings space [46, 59] and suggested that those could represent functional cell types (Fig. 2B). However, for more recent models with a Gaussian readout [51, 60] this structure is much less clear (Fig. 2C), although these models exhibit substantially better predictive performance (Fig. 2A). This vizualization difference was also evident in quantitative terms: Clustering the neuronal embeddings led to significantly more consistent clusters for the factorized readout than for the Gaussian readout (Fig. 2I; red vs. blue), independent of the number of clusters used.

We hypothesized that this difference might be caused by the $L_1$ regularization in earlier models, which could reveal the structure in neuronal function better and lead to more consistent embeddings. Indeed, the weight distribution for the factorized model was much sparser compared to the Gaussian readout (Fig. 2G). Increasing the $L_1$ regularization indeed resulted in more structured embeddings – at least qualitatively (Fig. 2D,E) and in terms of the distribution of embeddings (Fig. 2G; pink and green). It also came with a drop in predictive performance comparable to that of the factorized readout (Fig. 2A). However, somewhat surprisingly, it did not lead to more consistent cluster assignments (Fig. 2I; pink and green). This result suggests that the structure that emerges for the Gaussian readout, arises somewhat trivially by very aggressively forcing weights to be zero due to the strong L1 penalty, but this does not happen in a consistent way across models, therefore not improving the consistency

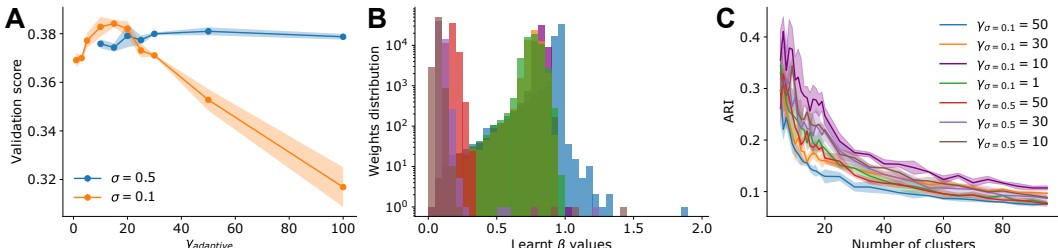

Figure 3: **Selection of adaptive regularization hyperparameters. A:** Validation correlation of model with $\sigma = 0.1$, $\sigma = 0.5$ and different overall regularization strength. For smaller $\sigma$ the performance decreases more fast. **B:** Distribution of $\beta$ values learnt. As expected, for smaller $\sigma$ the distribution is closer to $LogNorm$ and as we regularize both embeddings and $\beta$-s, the learnt $\beta$-s are smaller for bigger $\sigma$. This also explains, why the performance in panel A decreases slower, as the model is less regularized. **C:** ARI for different hyperparameters. We see that for $\sigma = 0.5$ ARI for $\gamma = 10$ and $\gamma = 30$ are mostly equivalent and $\gamma = 50$ is slightly worse. The ARIs trend is more visible on $\sigma = 0.1$. It seems like the better the predictive performance, the better is the ARI curve. As the performances for $\gamma = 10$, $\gamma = 15$, and $\gamma = 20$ are identical within std, we choose the least regularized model to introduce less bias.

of embeddings across runs. Note that the factorized readout is always $L_1$-regularized – it does not work without it [28] and is very sensitive to the choice of regularization strength (Appendix A.3). This is also why we show only a single gamma value in the table.

**Adaptive regularization improves consistency of embeddings while maintaining predictive performance.** We next asked what could explain the difference in embedding consistency between factorized and Gaussian readout. One important difference is that the factorized readout jointly regularizes the mask and the neuronal weight. As a consequence, the factorized readout could reduce the L1 penalty on the neuron weights by increasing the size of its receptive field mask (Fig. 2H). As a consequence, the neuronal weights may not be regularized equally.

Motivated by this observation, we devised a new adaptive regularization strategy for the Gaussian readout, where the strength of the $L_1$ regularizer is adjustable for each neuron:

$$L_1^{\text{readout}} = \gamma \sum_{n=0}^{N} \beta_n ||\mathbf{w}_n||_1 \tag{2}$$

Here, $\gamma$ controls the overall strength of regularization as before. The variables $\beta_n$ are learned coefficients for neuron $n$, onto which we imposed a log-normal hyperprior $p(\beta_n) \sim \text{LogNormal}(1, \sigma)$, achieved by adding the according loss term $L_\beta = 1/\sigma^2 \sum_{n=0}^{N} \log(\beta_n)$.

This adaptive regularization procedure introduces an additional hyperparameter, $\sigma$, that controls how far each neuron's regularization strength is allowed to deviate from the overall average controlled by $\gamma$. We found that for relatively large $\sigma = 0.5$, the model essentially learns to down-regulate all neurons' weights (Fig. 3B) and falls back to a very mildly regularized mode countering the effect of increasing $\gamma$ (Fig. 3A; blue). When constraining the relative weights more closely around 1 ($\sigma = 0.1$), the model indeed learns to redistribute the regularization across neurons. The average weights remain closer to 1 (Fig. 3B) and changing the overall regularization strength $\gamma$ does have an effect (Fig. 3A).

The optimal model chosen by this adaptive regularization strategy ($\gamma = 10, \sigma = 0.1$) indeed performs well on all dimensions we explored: First, its predictive performance matches the state-of-the-art model with the Gaussian readout (Fig. 2A). Second, its learned neuronal embeddings are more consistent that those of both factorized and uniformly regularized Gaussian readout (Fig. 2I; purple).

**Pruning models post-hoc improves consistency of neuronal clusters.** We saw that the consistency of neuronal clusters could be improved by adaptive regularization of the readout. However, in absolute terms the consistency is still not very high. One potential reason for this could be that deep models are typically highly overparameterized [34]. As a result, several of the feature maps of the core

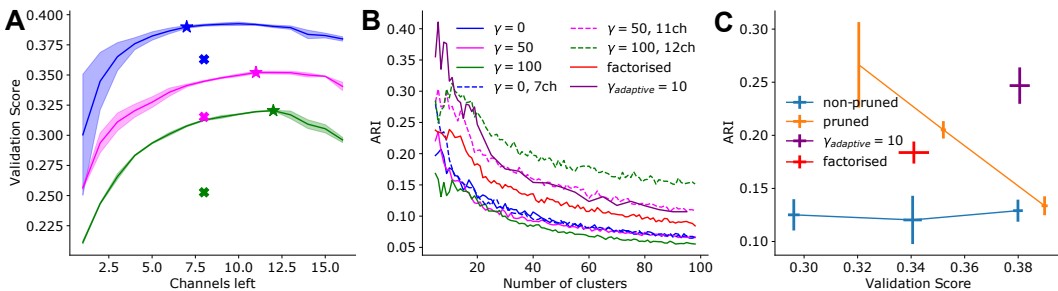

Figure 4: **Pruning. A**: Validation score of the pruned models with different regularization. Lines: average across three models. Shaded areas: standard deviation across three models. Colors: model type and regularization (see legend in B). Stars: selected model. Crosses: non-pruned model with 8 channels. **B**: ARI for non-pruned and models selected after pruning (stars from panel B). Pruning consistently improves consistency of clustering measured by ARI. Adaptive regularization readout pruning is shown in A.11. **C**: ARI-performance trade-off (20 clusters). Ideally we want high ARI and high performance.

could be redundant, resulting in multiple ways to represent the same neuronal function. In this case, two neurons with the same function could have highly different embeddings. We therefore now investigate whether pruning trained networks could lead to more consistent neuronal embeddings across model fits. We took the following approach to pruning models:

1. Train a model as usual.
2. In the trained model, mask one core output convolutional channel and compute performance.
3. Remove the channel for which masking decreases model performance the least.
4. Finetune the model on the reduced core.
5. Repeat steps 2–4 until only one channel is left.

In agreement with prior work on pruning neural nets [34], we observe a slight performance improvement during first pruning iterations and find that 1/3 to 1/2 of the convolutional channels can be removed without affecting performance (Fig. 4A). We selected the smallest number of channels before the performance starts to decrease. We also observed that for the more regularized models more channels should be kept after pruning (Fig. 4A), which makes sense as severe $L_1$ regularization is also a feature selection mechanism. Pruning noticeably improved the consistency of clustering neurons for models with the Gaussian readout (Fig. 4B), but interestingly only for the regularized models. However, this improvement in consistency still comes at the cost of predictive performance. Overall, the (non-pruned) model with the adaptive readout achieves the best consistency–performance trade-off (Fig. 4C, purple).

**Neuronal tuning properties.**   To explore to how improving reproducibility of neural embeddings impact neurons' actual behavior, we examined the predicted neurons' tuning with respect to four known properties of V1 neurons: orientation tuning, phase invariance, surround suppression and cross-orientation inhibition (see Section 4 for details). We chose these properties because they are well-established and widely studied nonlinear phenomena of V1 neurons [12, 17]. We performed in-silico experiments showing classical grating-based stimuli and computed neuronal tuning curves from the predicted model responses. Note that the neurons have not seen any of these stimuli during the in-vivo experiment. Hence, our models have not been explicitly trained on these stimuli. Thus, we cannot compare these predictions against ground truth. However, we can evaluate qualitatively whether the predicted tuning curves reproduce what is known from biology and we can quantify how consistently these phenomena are expressed across predictive models that just differ in their initialization (in an ideal world tuning curves would be identical across seeds).

We found that different readouts gave rise to substantially different tuning properties for the same neurons (Fig. 5, top row; one example neuron's tuning curves). Strong uniform regularization of Gaussian readouts ($\gamma = 50$ and $\gamma = 100$) made neuron's responses highly biased towards their mean response, while the non-regularized Gaussian model exhibited the biggest responses amplitude,

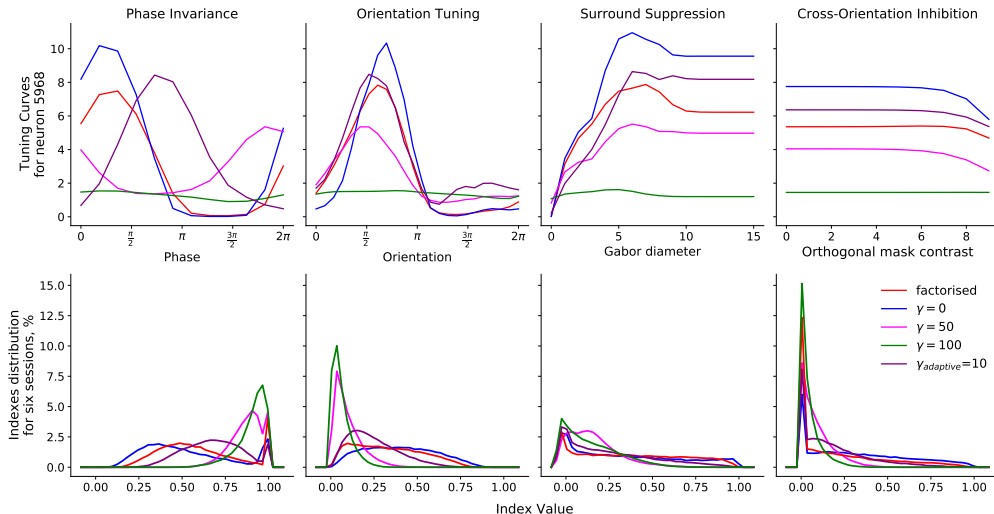

Figure 5: **Impact of regularization on neurons' tuning properties.** In silico analysis of neural tuning properties **Top:** Example of tuning curves for a neuron well-predicted by the model. **Bottom:** Population distribution of tuning indices across neurons.

followed by the adaptively regularized Gaussian and the factorized model, which both exhibit moderate shrinkage effects.

For each of the four response properties, we computed a tuning index that measures the degree of modulation. As tuning indices measure the effect size, tuning curves with small amplitude and high baseline lead to high phase invariance, and low orientation tuning, surround suppression, and cross-orientation inhibition tuning indices (distributions in Fig. 5 bottom). Non-regularized Gaussian and factorized models exhibited similar distributions. This result suggests that even though regularization leads to sparser and less overfitted embeddings, it also removes important functional properties due to reduced expressive power. The adaptively regularized models stayed in between of the non-regularized and highly regularized models, as it puts different regularization weights across various neurons, effectively combining both modes. Thus, it keeps the overall regularization coefficient $\gamma$ low, offering a good combination of both viable tuning properties and preserved consistent clustering.

## 6 Discussion

We performed several important reproducibility checks for the commonly used readout mechanisms of predictive models of the visual cortex. We found that the older factorized readout led to a more structured embedding space, more consistent neuronal clusters and more reproducible in-silico tuning

| Readout | $\gamma$ | Response Correlation ↑ | Cross-Ori. Inhibition ↓ | Phase Invar. ↓ | Orientation Tuning ↓ | Surround Suppr. ↓ |
|---|---|---|---|---|---|---|
| Factorized | 0.003 | 0.81 | **0.52** | **0.54** | **0.50** | **0.78** |
| Gaussian | 0 | 0.76 | 0.72 | 0.69 | 0.69 | 0.85 |
| | 50 | **0.86** | 1.10 | 0.92 | 1.03 | 1.26 |
| | 100 | 0.82 | 0.91 | 0.83 | 0.96 | 1.04 |
| Gaussian adaptive | 10 | 0.85 | 0.81 | 1.02 | 0.81 | 0.94 |
| Gaussian pruned | 0 | 0.85 | 0.72 | 0.71 | 0.79 | 0.92 |
| | 50 | **0.86** | 0.91 | 0.91 | 0.99 | 0.95 |
| | 100 | 0.83 | 0.96 | 0.82 | 1.02 | 0.99 |

Table 1: **Consistency of model predictions and tuning properties across model fits.** We compared pairwise correlations of response predictions across three model fits (↑: higher is better) and the normalized mean absolute error (NMAE; Sec. 4) across tuning indeces (↓: lower is better).

curves than the more recent, performance-optimized Gaussian readout. The structure of the embedding space was primarily related to the $L_1$ regularization of the weights. By equipping the more recent Gaussian readout with a novel, adaptive $L_1$ regularization, we could recover a similarly structured embedding space and achieve more consistent neuronal clusters without sacrificing predictive performance. However, all of the models with a regularized Gaussian readout exhibited strongly biased neuronal tuning curves, calling into question how faithfully they can represent neuronal tuning beyond experimentally verified properties such as maximally exciting stimuli [37, 47, 62, 63, 68]. Rigorous, biologically meaningful evaluation remains an open question, but the presence of non-linear phenomena (tuning curves) is crucial. While current methods focus on explained variance/correlation, models with similar scores can differ in explainability [53]. To our knowledge, we conducted state-of-the-art evaluations for biological meaning by testing the tuning curves.

An important goal of models is that they should be resistant to small hyperparameters changes and different initial conditions in order to lead to meaningful biological conclusions. The comparably low consistency of state-of-the-art models across seeds suggests that future research is required to improve the consistency of models, potentially by improving identifiability or improved training paradigms that find robust solutions.

One limitation of our work is that we focused on one core architecture (rotation-equivariant CNN). We do not have a reason to believe that other core architectures would behave fundamentally differently. One sanity check with a regular CNN core (Appendix A.6) yielded similar results. Whether the same holds for Vision Transformer cores [64] remains to be checked.

In conclusion, our work (1) raises the important question how reproducible current neuronal predictive models are; (2) suggests a framework to assess model consistency along orthogonal dimensions (clustering of neuronal embeddings, predicted responses and tuning curves); (3) provides a technique to improve the consistency of the clusters with keeping the predictive performance high.

## Acknowledgements

We would like to thank Konstantin F. Willeke, Dmitry Kobak and Ivan Ustyuzhaninov for discussions, as well as Rasmus Steinkamp and the GWDG team for the technical support. MFB thanks the International Max Planck Research School for Intelligent Systems (IMPRS-IS).

This work was supported by the Deutsche Forschungsgemeinschaft (DFG, German Research Foundation) – project ID 432680300 (SFB 1456, project B05) – and by the European Research Council (ERC) under the European Union's Horizon 2020 research and innovation programme (grant agreement number 101041669). Computing time was made available on the high-performance computers HLRN-IV at GWDG at the NHR Center NHR@Göttingen. The center is jointly supported by the Federal Ministry of Education and Research and the state governments participating in the NHR (www.nhr-verein.de/unsere-partner). FHS is supported by the German Federal Ministry of Education and Research (BMBF) via the Collaborative Research in Computational Neuroscience (CRCNS) (FKZ 01GQ2107).

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

# A  Appendix

## A.1  Model config and Training pipeline

We stayed close to the Willeke et al. [60] benchmark with list of the following changes. **For the model**, we used rotation equivariant model with 16 *hidden_channels* and 8 rotations per channel, which resulted in the total readout dimensionality of 128. *grid_mean_predictor* was turned-off, *input_kern* was 13 and *hidden_kern* was 5, to make models closer to Ecker et al. [33] and Ustyuzhaninov et al. [59]. *gamma_input*, corresponding to the input Laplace smoothing was set to 500 and *gamma_hidden*, applying Laplace smoothing over the first layer of convolutuonal kernels was set to 500,000, *depth_separable=False* **For the training**, we also used early stopping and learning rate scheduling with 4 steps. The only differences were that we used *batch_size*=256 instead of 128 and initial learning rate of 0.005 instead of 0.009.

## A.2  Parameters of optimal gabor and tuning curves sweep

For the optimal Gabor search, we followed Burg et al. [53] and generated the Gabors with 6 contrasts steps and 8 sizes steps. The maximum input value for the stimuli in the dataset was 1.75, which we used for the maximum contrast and the minimal contrast we set to 1% of the maximum. The stimuli diameter sizes were between 4 and 25. Minimum spacial frequency was $1.3^{-1} \cdot 1.3^i, i = 0, 1, ..., 10$.

For **orientation tuning** 24 orientations equidistantly partitioning the interval between 0 and $\pi$ were used with 8 phases and maximum response acrross 8 phases was taken for each orientation.

For **phase invariance** we created stimuli for twelve equidistant phases between 0 and $2\pi$.

For **surround suppression** we used the following stimulus creation parameters

```
p = {
    'total_size' : 40,
    'min_size_center' : 0.05,
    'num_sizes_center' : 15,
    'size_center_increment' : 1.23859,
    'min_size_surround' : 0.05,
    'num_sizes_surround' : 15,
    'size_surround_increment': 1.23859,
    'min_contrast_center' : 2,
    'num_contrasts_center' : 1,
    'contrast_center_increment' : 1,
    'min_contrast_surround' : 2,
    'num_contrasts_surround' : 1,
    'contrast_surround_increment' : 1
}
```

For **cross-orientation inhibition** 9 contrasts and 8 phases were used to create the combinations of the optimal Gabor and its orthogonal version.

## A.3  Choice of regularization for the factorized readout

For factorized readouts severe $L_1$ regularization is design necessity because mask and readout are regularized jointly. The idea of the mask is to learn the receptive field, ideally 1 'pixel' only, so the strong regularization is needed and tightly connected with the performance. We select the regularization based on the best performance - $\gamma = 0.003$ (see Fig. 6).

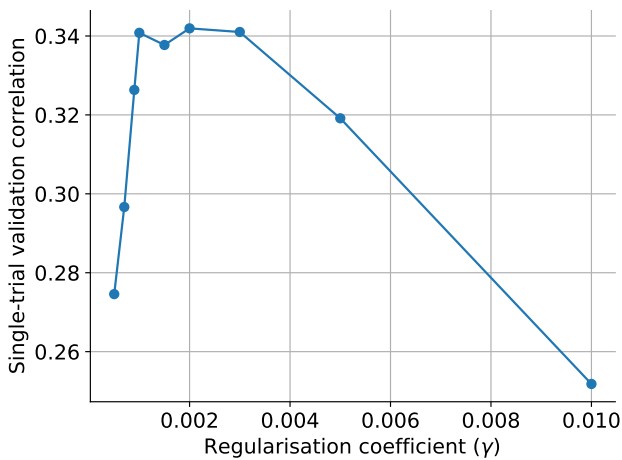

Figure 6: Regularization strength impact on performance for factorized readouts

### A.4 ARI stability - 1 model and 3 seeds for k-means

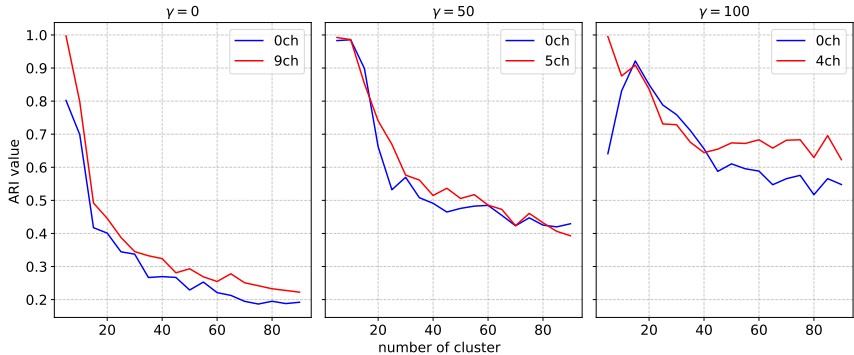

Figure 7: Here for the original and pruned models we took the model trained in 1 seed ($seed = 101$) and performed the clustering using 3 different seeds in k-Means and computed ARI across different partitions. As k-Means is initialisation dependent we can see that in case of non-clear clusters even on the same vectors the ARI values could be very low. As we do not know the true amount of clusters, we tried clusters in range [5, 90] with $step = 5$.

### A.5 Selection of channels during pruning

To select the amount of channels, we train same models with 3 different starting seeds and estimate the mean $bar\rho$ and $\sigma_\rho$ of performance on each pruning stage. accounting for *std* using the following logic: $\bar{\rho}_i - \sigma_{\rho_i} <= \bar{\rho}_{i+1} - \sigma_{\rho_{i+1}}$ where $i$ is the amount of channels pruned.

### A.6 Motivation for rotation equivariant core and non-equivariant control

The motivation behind the rotation-equivariant core is that V1 neurons are orientation selective and neurons with all preferred orientations exist. Thus, any feature is likely to be extracted in multiple rotated versions [33, 46, 59]. To control that our results are not specific to the rotation-equivariant core, we performed a few control experiments with a non-rotation-equivariant CNN core (Fig. A.6): The general pattern of results is the same: the adaptive regularization improves consistency of clustering and so does pruning feature maps.

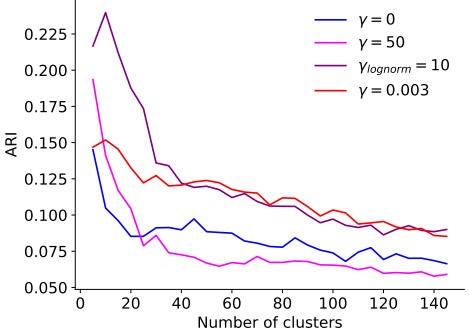

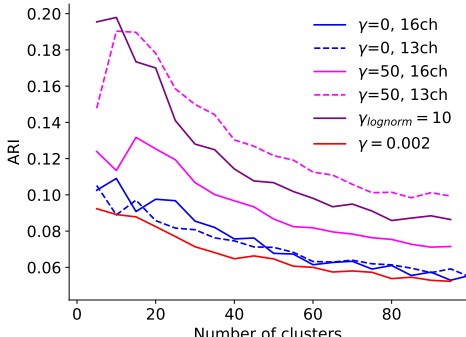

Figure 8: **Non**-rotation equivariant CNN with 128 channels ((rotation equivariant core with 16 channels and 8 rotations would have "128" channels). Adaptive regularization outperforms both regularized Gaussian and factorized models in terms of ARI. In both plots - red is factorised, purple is adaptive Gaussian, blue is non-regularized Gaussian, magenta is regularized Gaussian.

Figure 9: Pruned **non**-rotation equivariant CNN core with 16 channels (we decreased the amount of channels to make the computations feasible). We see that pruning improves ARI and adaptive model is still better then regularized Gaussian and factorized models. Though, changing the dimensionality might require additional research to scale regularization strength.

### A.7 Table with correlations for shifter turned on

| Readout | $\gamma$ | Responses Correlation ↑ |
|---|---|---|
| Gaussian | 0 | 0.65 |
| | 50 | 0.80 |
| | 100 | 0.75 |
| Gaussian pruned | 0 | 0.80 |
| | 50 | 0.81 |
| | 100 | 0.77 |

Table 2: This table is same logic as Table 1 but here for the Gaussian readouts the *shifter* was not used. We can see that with *grid mean predictor* undefined the shifter actually improves the consistency of responces 1, while pruning still helps, especially for the non-regularized models. However, turning off the shifter causes some displacements in the receptive fields, probably in different directions. This effect is much more present, when *grid mean predictor* is on as it can compensate for the shifter displacement even more.

## A.8    Optimal Gabors

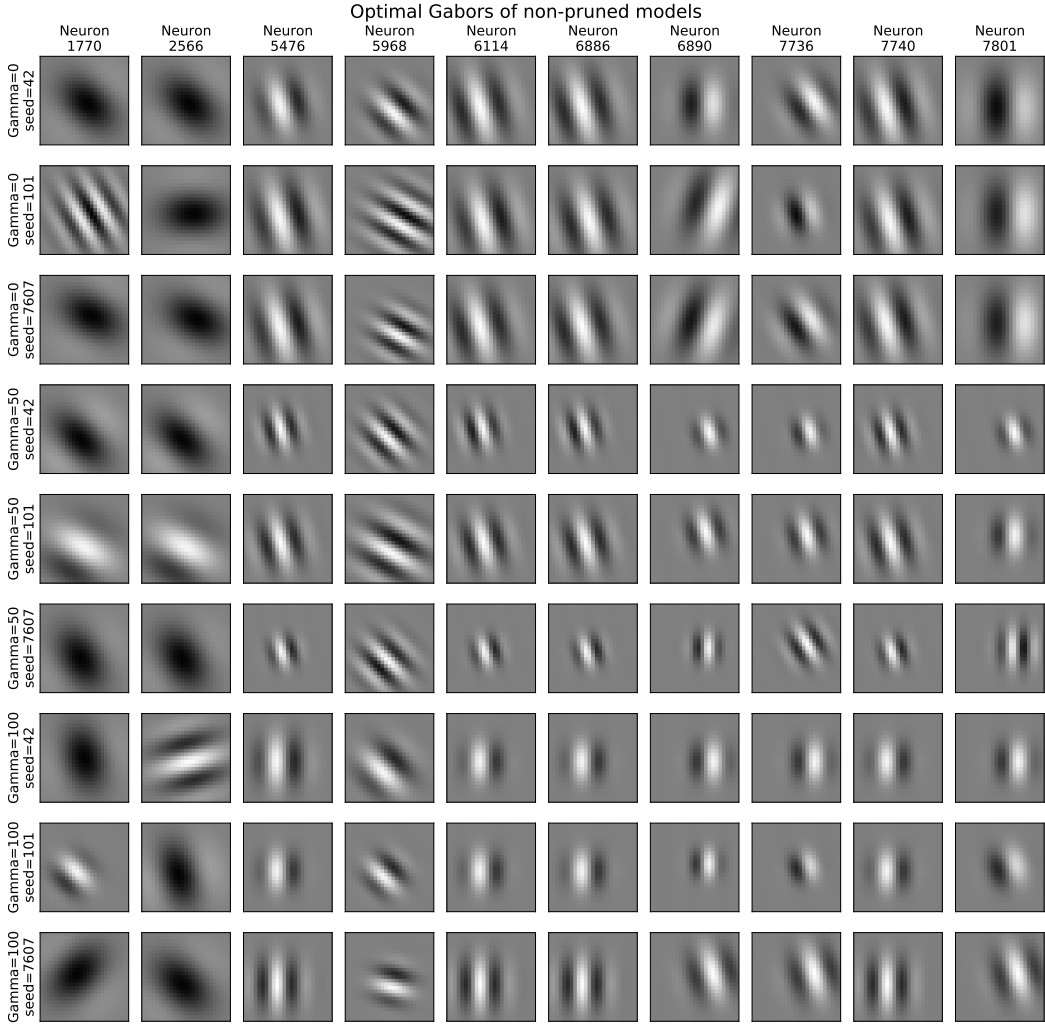

Figure 10: Optimal Gabors for non-pruned models. Columns are per neuron, rows are per model, for each of three regularization strengths three seeds were trained. We can see that the optimal Gabors selected are somewhat consistent, however, not ideally same. More regularized models also tend to select Gabors with smaller sizes.

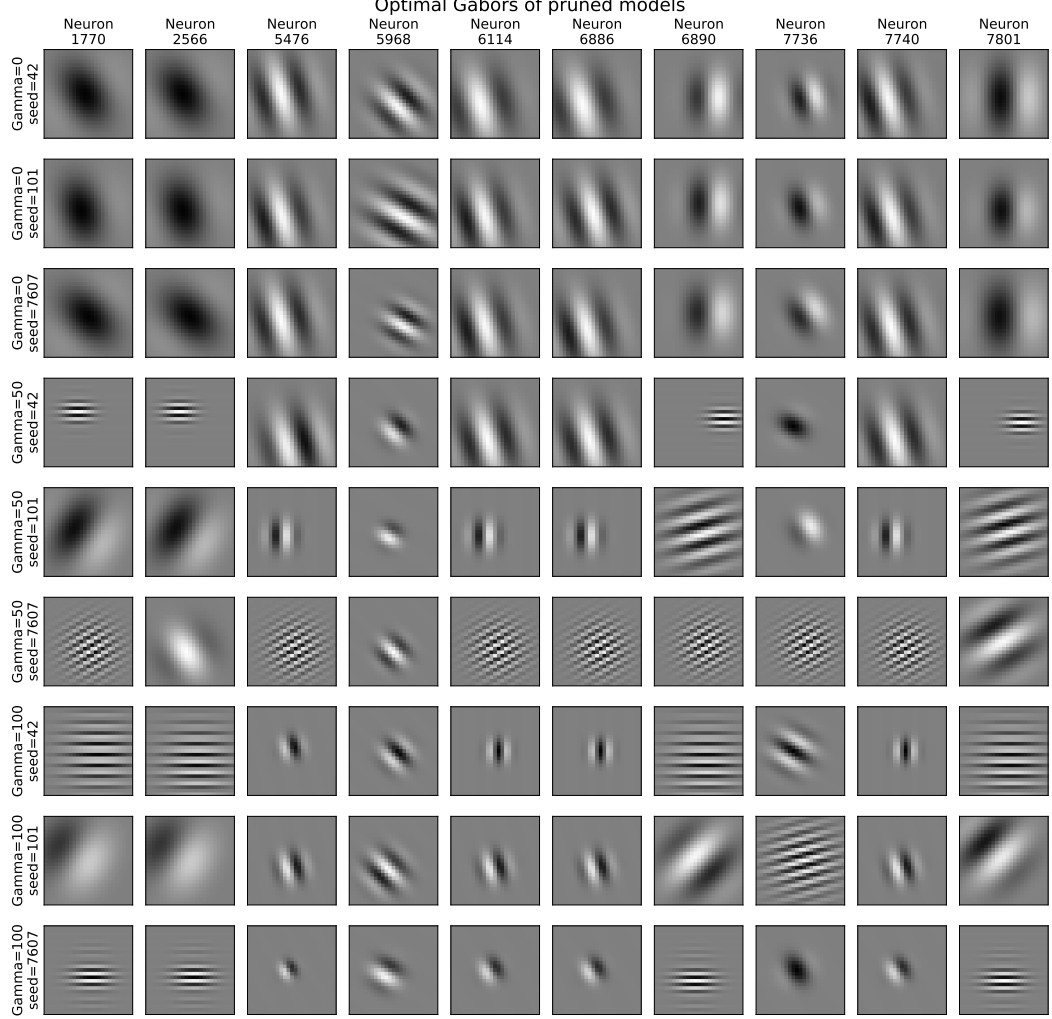

Figure 11: Optimal Gabors for pruned models. Columns are per neuron, rows are per model, for each of three regularization strengths three seeds were trained. Compared to Fig. 10 we see that for $\gamma = 0$ the model became much more consistent across seeds, suggesting that the model was originally overparametrised. Moreover, pruned $\gamma = 0$ models now select optimal Gabors, which are close to the non-pruned majority choice across models. While this is not true for the more regularized models, they now disagree more between seeds and tend to choose Gabors with higher-frequencies, which is not a biologically plausible choice. This is probably happening due to too severe regularization. However, this explains why there is a bigger differences between the tuning indexes for more regularized models. The indexes are computed on modifications of the optimal Gabors, and if the Gabors are different the responses and indexes would be different as well.

## A.9 Other clustering consistency metrics

ARI is the most popular metric for pairwise cluster comparisons [19], which we think is more appropriate than set-based methods. To ensure we are not missing anything, we also considered alternative metrics such as Fowlkes-Mallows, completeness, homogeneity, and v-measure (a particular case of mutual information) and repeated the analysis. The results show the same ordering of methods A.9. Do the metrics reflect biological significance? We reasoned that if neuronal embeddings determine the response function of a neuron in the model, then two neurons that have similar

embeddings under one instance of a model, should also have similar embeddings under another instance of this model. Therefore, measuring similarity of clustering is a reasonable proxy.

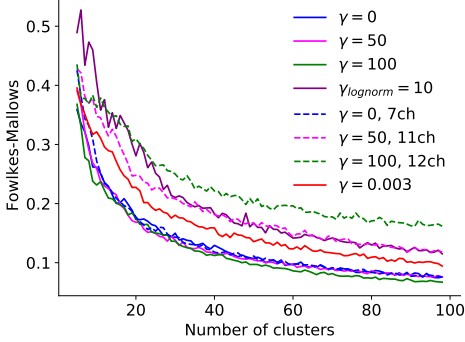

Figure 12: Fowlkes-Mallows index for estimating the cluster consistency. It shows same trends as ARI. In both plots - red is factorised, purple is adaptive Gaussian, blue is non-regularized Gaussian, magenta and green are regularized Gaussian.

Figure 13: V-measure index for estimating the cluster consistency. The models order is same as ARI, though it is biased towards bigger amount of clusters due to the set-based nature.

## A.10 t-SNE reliability

Please note that we used t-SNE only for visualization purposes to show the presence of the density modes in the data. While t-SNE plots are indeed not sufficient to make any conclusions about the data, many papers showed that they are a good tool for exploratory data analysis [72] and we additionally checked trustworthiness metric [18] after performing t-SNE. Trustworthiness shows higher scores for more regularized models with varying numbers of neighbors confirming the presence of the density modes (Fig. 14).

## A.11 Adaptive regularization pruning

It turns out that pruning does not further increase ARI for the adaptively regularized model (Fig. 15). Thus, regularizing neurons in a non-uniform fashion is already effective enough that additional pruning does not improve it. We speculate that this is because some noisy neurons become sufficiently regularized without pruning, while before such regularization was only achieved with pruning and regularization combined.

## A.12 Computational resources

All of the experiments were performed on a local infrastructure cluster with 8 NVIDIA RTX A5000 GPUs with 24Gb of memory each. 2 models could be trained on a single GPU simultaneously in $\approx$ 3 hours. Pruning experiment, without parallelization, would take a week. No extensive cpu resources are required. Optimal Gabor search, without parallelization takes around 12hours for Gaussian readout model and 3 days for the factorized readout model. We also pre-saved optimal Gabors in batches to improve the speed, which required $\approx$ 100Gb of memory.

## A.13 Broader impact

Our work is an important step towards obtaining a functional cell types taxonomy for the primary visual cortex neurons. Such classification can significantly improve our understanding of the brain and help on the way to understand the mechanisms and develop the treatment for different neurodegenerative disease.

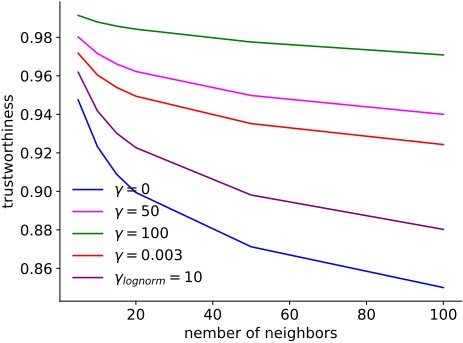

Figure 14: Trustworthiness of the 2 dimensional t-SNE projections. It shows higher scores for more regularized models, confirming the presence of the density modes. In both plots red is factorised, purple is adaptive Gaussian, blue is non-regularized Gaussian, magenta and green are regularized Gaussian.

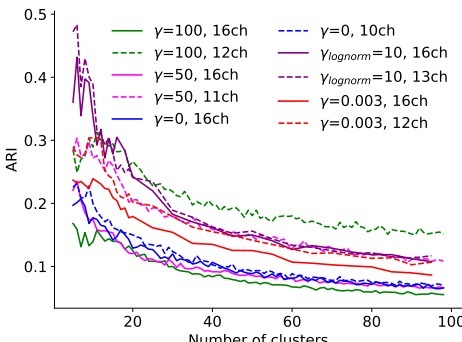

Figure 15: Pruning for rotation-equivariant modeling with adaptive regularization of Gaussian readout and factorized readouts.

