# OpenReview forum: "Reproducibility of predictive networks for mouse visual cortex"
_NeurIPS.cc/2024/Conference — NeurIPS 2024 spotlight_

### Official Review · Reviewer_iat5 · 2024-06-29

**Soundness:** 4
**Presentation:** 2
**Contribution:** 4
**Rating:** 7
**Confidence:** 3

**Summary:**

This work demonstrates that overparameterized neural network models, which have many non-unique solutions, can lead to inconsistencies in representing the mouse visual cortex. It suggests a novel approach called "adaptive regularization," where the regularization parameters in the loss term are learnable rather than fixed. The study also examines other approaches, such as normal regularization and pruning, and finds that these methods are beneficial for improving representation consistency. The proposed method significantly enhances consistency. This contribution is significant to the computational neuroscience community.

**Strengths:**

1. This work addresses a critical problem in computational neuroscience: reproducibility. It proposes solutions to mitigate this issue, which have significant impact on the computational neuroscience community.

2. The authors rigorously define the consistency of computational models and thoroughly examine how regularization strength affects neuronal properties.

**Weaknesses:**

1. The study is limited to only one type of model, the Rotational Equivariance CNN.

2. In the Methods section, the terms "Embedding" and "Mask" are not defined. Moreover, Lp and L1 are described as functions of model parameters, but no parameters are defined in the text. This lack of clarity poses an issue for understanding. For instance, Lp could be interpreted in two different ways: as a function of the core but not the readout, or as a function of both the core and the readout. The same ambiguity applies to L1.

3. There are two types of computational neuroscience models regarding their outputs. The first type is task-optimized models, whose outputs are task-related, such as object class, object representation, or action. These models are trained to perform downstream tasks, such as supervised object classification, reconstruction, or playing games [1,2,3]. The second type is neural response fitting models, whose outputs are predicted neural responses, and they are trained to predict these responses directly. However, the reasons why the authors chose response fitting models over task-optimized models are not mentioned.

Reference

[1] Performance-optimized hierarchical models predict neural responses in higher visual cortex

[2] Unsupervised neural network models of the ventral visual stream

[3] Using deep reinforcement learning to reveal how the brain encodes abstract state-space representations in high-dimensional environments

**Questions:**

1. Oother models that, despite lacking rotational equivariance, still demonstrate high capability in predicting neural responses measured by electrodes [1]. Why did the authors choose the Rotational Equivariance CNN? What was the motivation behind this choice?

2. Figure 2A shows the models' performances. However, what about the factorized readout with L1 regularization? Does the predictive performance decrease?

3. Why did the authors choose a log-normal prior for the learnable coefficient?

4. Why did the authors not examine adaptive regularization for factorized readout models? The implementation should be straightforward, similar to what was done for Gaussian.

5. Why does the factorized model have one γ in Table 1?


Reference

[1] https://www.brain-score.org/vision/

**Limitations:**

This study investigates the consistency of neuronal properties and the prediction performance of regularized models. However, the models used by the authors are trained to predict neural responses. There is another type of computational model that is trained to perform downstream tasks, with neural-like representations emerging from the training. Future work could consider these latter models.

---

> ### Author Rebuttal · Authors · 2024-08-07
>
> Thank you very much for your rigorous review. We highly appreciate your feedback, which helps us to improve our paper.
>
> Regarding the weaknesses you mention:
> * **“The study is limited to only one type of model, the Rotational Equivariance CNN”**
> The motivation behind the rotation-equivariant core is that V1 neurons are orientation selective and neurons with all preferred orientations exist. Thus, any feature is likely to be extracted in multiple rotated versions. The rotation-invariant framework allows to assess response features independent of the preferred orientation of a neuron [1,2]. Moreover, it does not lead to any sacrifice in performance compared to the classic CNNs reported in Sensorium 2022 competition [3]. We do not see a reason why our results should be specific to the rotation-equivariant core, though. To justify this intuition, we performed a few control experiments with a non-rotation-equivariant CNN core (Fig. 2 + 5 in attached PDF): The general pattern of results is the same: the adaptive regularization improves consistency of clustering and so does pruning feature maps. We hope these controls solidify your trust in our results.
>
> * **“In the Methods section, the terms "Embedding" and "Mask" are not defined.”**
> We will improve the clarity for “masks” and “embeddings” definitions. In fact, Fig. 1A and 2H are supposed to show it, but we realize it’s not entirely clear. We will add annotations to the figures and include a sentence in the text. Both are weights learned in the readout. The “mask” is a 2D matrix, selecting a “receptive field” from the latent space and the “embedding” is a linear vector representing a learned neuron function.
>
> * **“Moreover, Lp and L1 are described as functions of model parameters, but no parameters are defined in the text. This lack of clarity poses an issue for understanding. For instance, Lp could be interpreted in two different ways: as a function of the core but not the readout, or as a function of both the core and the readout. The same ambiguity applies to L1.”**
> We apologize for the lack of clarity and will improve the exposition. To clarify: The regression loss Lp is the main loss driving model fitting. It penalizes the difference between observed neuronal responses $r$ and model predictions $\hat r$. As the response predictions $\hat r$ are the model *outputs*, they are a function of both core and readout and hence guide optimization of *all* parameters. L1, in contrast, is only a function of the readout parameters, which we believe should be clear from its definition in line 133, which sums over mask $m$ and embeddings $w$ – both readout parameters (but admittedly not entirely clear as already discussed in the previous point).
>
> * **“The reasons why the authors chose response fitting models over task-optimized models are not mentioned.”**
> It’s true, we could also have used a core from a task optimized model and finetune the readout with it. However, (a) this would lose the rotation equivariance, (b) task-driven models are not great representations for mouse V1 [4] and (c) our main question is about the reproducibility of the learned representations, so we would need multiple versions of the task-trained model. This would (i) imply retraining the task-trained models several times, (ii) would raise lots of questions about which ones to use and (iii) how the specifics of the task training affect the results as well as (iv) whether these representations should be fine-tuned when training the readout or not. While all of these are valid and interesting questions, we opted for focusing on data-driven models for this study and deferring an analysis of task-driven models to future work.
>
> As for the questions:
>
> * **Why did the authors choose the Rotational Equivariance CNN?** Answered above
> * **“Figure 2A shows the models' performances. However, what about the factorized readout with L1 regularization? Does the predictive performance decrease?” and “Why does the factorized model have one γ in Table 1?”**
> Note that the factorized readout is always L1-regularized – it does not work without it [5] and is very sensitive to the choice of regularization strength $\gamma$ (Appendix, Fig. 6). This is also why we show only a single gamma value in the table.
>
> * **“Why did the authors choose a log-normal prior for the learnable coefficient?”**
> Using this prior we separate the overall strength of regularization ($\gamma$) from the weight $\beta_n$ of each individual neuron. In this parameterization defined in Eq. (2) we want the weights $\beta_n$ to be non-negative and 1 on average, which is what a lognormal prior achieves. We also experimented with restricting coefficients to be positive and using L1 or L2 penalties, but this did not work well because they pushed many coefficients to zero, effectively avoiding regularization.
>
> * **“Why did the authors not examine adaptive regularization for factorized readout models? The implementation should be straightforward, similar to what was done for Gaussian.”**
> The factorized readouts are always (implicitly) adaptively regularized. This insight is what inspired us to develop the adaptive regularization approach. As Figure 2H illustrates, the factorization allows moving weight between the mask and the embedding. By changing the size of the mask, one can effectively change the L1 penalty of the embedding vector. As the Gaussian readout always selects exactly one location, it does not have this degree of freedom.
>
> We hope these comments shed some light and clarified our motivation, technical and design choices.
>
> [1] Ustyuzhaninov et al. (2022) doi: 10.1101/2022.02.10.479884
> [2] Ecker et al. (2019) doi: 10.48550/arXiv.1809.10504
> [3] Willeke et al. (2022) doi: 10.48550/arXiv.2206.08666
> [4] Cadena et al. (2019) NeurIPS 2019 Workshop Neuro AI
> [5] Klindt et al. (2017) doi: 10.48550/arXiv.1711.02653

---

> ### Comment · Area_Chair_EnMd · 2024-08-12
> **Gentle reminder to respond to rebuttal**
>
> Dear Reviewer iat5,
>
> As the reviewer-author discussion closes soon (Nov 13 11:59 pm AoE), please let us know if the author rebuttal addressed your questions and whether your maintain or modify your original score.
>
> Gratefully,
>
> AC

---

### Official Review · Reviewer_8oHp · 2024-07-11

**Soundness:** 4
**Presentation:** 3
**Contribution:** 3
**Rating:** 7
**Confidence:** 3

**Summary:**

The authors present a systematic investigation of the use of deep neural network fits to biological neurons as the basis for neuron cell type classification. The authors explore how various factors such as regularization and model pruning can influence both the predictive model fit and the consistency of the neural clustering.

**Strengths:**

This paper presents a well-motivated, calculated set of experiments for evaluating how well deep neural networks can be used as models of mouse visual cortex. With increasing interest in this area and increasingly bold claims, work like this is a breath of fresh air. The controlled experiments and attention to detail provides compelling evidence that though DNNs may be our best working computational model of visual cortex, the are likely not a global optimum.

**Weaknesses:**

Overall, the biggest weakness I think is some lack of explanation. As someone who is very familiar with work in computational neuroscience but not a neuroscientist myself, there were a few things that felt under-explained and it wasn't clear if it was because I'm not the correct audience or because the authors did not provide enough context in the text. The authors do a good job of providing clear motivation for some of the fundamentals (L24-53) but then leave out context when things get a little more technical specialized (i.e. comparisons between readout mechanisms, significance of biological properties and tuning indices, etc) as a result it feels unclear what the reader is expected to know before reading.

- The font in the figures is a little unprofessional. Comic sans? In this economy?
- Figure 2G is hard to parse. I recommend splitting up the histograms
- The intuitive explanation of the factored vs gaussian vs adaptive readouts is not super clear to me. It might be worthwhile to spend more time on it, especially to motivate your proposed adaptive mechanism.
- minor typos ("initialiation" in L183)
- L216 is nonsensical. I think some words are out of order
- Confidence intervals in Table 1 would be nice to gauge significance. Also, the table format makes it a little hard to visualize. Perhaps a graphical representation would be better.

**Questions:**

- Why is the adaptive model not pruned in the experiments shown in fig 4? I feel like it makes sense to include it in the pruning experiments if possible for consistent comparison across experiments.

- I think more explanation of table 1 is warranted. First, what is the ideal case? It's not clear to me that the optimal computational model has neurons with precise tuning curves for these chosen functional biological properties. If this is obvious to the authors, then perhaps there should be explicit motivation in the text. Why these biological properties and not others? Do biological neurons in HVC demonstrate each of these properties robustly (and not others)? Is it fair to expect that neurons in silica should each individually exhibit these properties? Second, the difference in NMAE of tuning indices between the factorized and the gaussian readouts seems very large. Large enough that I would expect more discussion of it in the text. How big of a difference in score is this really, is there something that can put it in context? If the biological property fits are a big deal, then is this a nail in the coffin for gaussian readout? Is gaussian readout a hack for improving overall fit while sacrificing something more important? Or is the functional biological properties just a cool trick that would be nice to have but not essential.

**Limitations:**

Limitations are clear.

---

> ### Author Rebuttal · Authors · 2024-08-07
>
> Thank you for appreciating our paper stating that “with increasing interest in this area and increasingly bold claims, work like this is a breath of fresh air.” Regarding your questions:
>
> * **“Why is the adaptive model not pruned in the experiments shown in fig 4? I feel like it makes sense to include it in the pruning experiments if possible for consistent comparison across experiments.”** Thank you for bringing this up. As you suggested, we now pruned the adaptively regularized model as well (see Fig. 3 in attached PDF). It turns out that pruning does not further increase ARI for the adaptively regularized model. Thus, regularizing neurons in a non-uniform fashion is already effective enough that additional pruning does not improve it. We speculate that this is because some noisy neurons become sufficiently regularized without pruning, while before such regularization was only achieved with pruning and regularization combined.
>
> * **“More explanation of table 1 is warranted. First, what is the ideal case?”** Ideally, a model would accurately predict the measured neural responses (response correlation across models should be high - ideally one (1) and result in the same biological phenomena. However, we cannot compare biological phenomena against the ground truth, as they are not contained in the dataset. We can, though, quantify how consistently these phenomena are contained across predictive models that just differ in their initialization seed, and would expect a perfect match (low NMAE, ideally 0).\
> \
> *“It's not clear to me that the optimal computational model has neurons with precise tuning curves for these chosen functional biological properties. [...] Why these biological properties and not others? Do biological neurons in HVC demonstrate each of these properties robustly (and not others)? Is it fair to expect that neurons in silico should each individually exhibit these properties?”*\
> Note that we train the models to predict the activity of neurons in the brain and analyze only the output neurons (i.e. the “digital twins” of the real neurons). Thus, each in-silico output neuron should behave as its corresponding real neuron. We chose the biological properties because they are well-established and widely studied nonlinear phenomena of V1 neurons ([1,2]  and many more) and our dataset is from V1 only (i.e. no HVAs). Note also that we do not expect all neurons – neither in-vivo nor in-silico – to exhibit all these properties. The indices we measure merely quantify *to what degree* they exhibit them (Fig 5, original manuscript) – and all model instances should consistently come to the same answer (Table 1, original manuscript).\
> \
> *“Second, the difference in NMAE of tuning indices between the factorized and the gaussian readouts seems very large [...] is there something that can put it in context? If the biological property fits are a big deal, then is this a nail in the coffin for gaussian readout? Is gaussian readout a hack for improving overall fit while sacrificing something more important? Or is the functional biological properties just a cool trick that would be nice to have but not essential.”*\
> No, it is not a nail in the coffin. It shows that the factorized readout produces more consistent indices across model runs. Whether these indices are more *accurate* is not captured by these numbers. That question can be resolved quantitatively only by an actual experiment where one measures real tuning curves. However, Fig. 5 in the original manuscript suggests qualitatively that the tuning curves by non-regularized Gaussian, factorized and adaptive readouts are more consistent with what we know from biology than regularized Gaussian readouts.
>
>
> In addition, we will include your other suggestions into the final version of our manuscript substantially improving the presentation of our results.
>
> [1] Cavanaugh et al. (2002) doi: 10.1152/jn.00692.2001
> [2] Busse et al. (2009) doi: 10.1016/j.neuron.2009.11.004

---

> > ### Comment · Reviewer_8oHp · 2024-08-12
> > **Response**
> >
> > Thank you for the authors response. I will update my review accordingly.

---

### Official Review · Reviewer_mU1L · 2024-07-15

**Soundness:** 3
**Presentation:** 3
**Contribution:** 4
**Rating:** 8
**Confidence:** 4

**Summary:**

This work studies models trained to predict the responses of neurons in visual cortex. The model has a shared multilayer network core followed by a final layer which maps the core features into individual neuron responses. The key question that this paper asks is, How reproducible are the individual neuron properties and "cell-type" clusterings inferred by such techniques? The main conclusion is that sparsity-inducing regularization is important for finding consistent cell properties. Pruning certain channels in the core is also shown to improve consistency.

**Strengths:**

Consistency and reliability of these methods for understanding visual cortical processing is important for the scientific interpretation of these models. The design of the approach seems sound. A number of metrics are used, both scores as well as response properties used in neuroscience.

**Weaknesses:**

* I found the comic sans font and cartoon diagram style distracting in Figs 1-4.
* It isn't clear that much is gained from the t-SNE plots in Fig 2. Some evidence of different clusters, yes, but it's known that these kind of plots can be deceiving.
* How do you pick the optimal regularization parameters? By what metric/cross-validation procedure? (Line 242 gives $\gamma=10, \sigma=0.1$) Validation performance is reported but I'm unclear of the splitting procedure and whether there was a train/valid/test 3-way split or not.
* Minor points are made in "questions" section

**Questions:**

* Line 35 "well described" -> "well-described" typo
* Line 62 "We address the identifiability problem, by proposing" typo, remove comma
* Fig 1 caption "sampling receptive field positions form a Gaussian" typo
* Is the clustering shown in Fig 1B truly "rotation invariant" or related to cyclic permutations of the vector indices? It is unclear from the text. The figure does not depict a rotation (i.e. action by an orthogonal matrix), since the orbit is elliptical rather than circular and it is not centered at the origin.
* Fig 3A has $\sigma=01$ which should be $\sigma=0.1$
* Fig 3B axis "amount of such weights" sounds strange

**Limitations:**

The authors mention that their work is limited to a particular type of core architecture. It's also limited to being applied to just one dataset, whereas open data in other animals (or collected by other research groups, e.g. the Allen Institute's work) are available. I am not suggesting the authors do this in their revision but that they mention it as a limitation.

---

> ### Author Rebuttal · Authors · 2024-08-07
>
> Thanks a lot for your time and effort for the review. We are very happy to see your feedback and high evaluation of our work. We will fix the typos, clarify details and address the style comments in the final version.
>
> You basically raise two main concerns about the t-SNE plot in Fig. 2 and the selection of hyperparameters in regularization:
>
> * **It isn't clear that much is gained from the t-SNE plots in Fig 2. Some evidence of different clusters, yes, but it's known that these kind of plots can be deceiving.** Please note that we used t-SNE only for visualization purposes to show the presence of the density modes in the data. While t-SNE plots are indeed not sufficient to make any conclusions about the data, many papers showed that they are a good tool for exploratory data analysis [1] and we additionally checked `trustworthiness` metric [2] after performing t-SNE. Trustworthiness shows higher scores for more regularized models with varying numbers of neighbors (see Fig. 1 in attached PDF) confirming the presence of the density modes.
> * **“How do you pick the optimal regularization parameters? By what metric/cross-validation procedure? Validation performance is reported but I'm unclear of the splitting procedure and whether there was a train/valid/test 3-way split or not.”** We do not use a test set but report performance on the validation set. However, that’s not a problem because (a) predictive performance is not our primary target and (b) in all cases where we select a single model from multiple (Fig. 3A, Fig. 4A, Fig. 6) the performance of the models left and right of the one selected are virtually the same (difference < 0.01), so none of the results depends on this particular model. Hence, we use the original train/val splits of the Sensorium 2022 competition as the additional effort of doing another data split is not warranted.\
> For Gaussian readouts $\gamma = 0, 50, 100$ were chosen arbitrarily, as there is no globally “optimum” parameter, as higher regularization leads to better ARI but hurts predictive performance. The factorized readout is regularization chosen to maximize performance on the validation set, however, this model simply does not learn if the regularization is not correctly tuned (see Fig 6 in the appendix or [4]).
>
> As for the questions:
> * **“Is the clustering shown in Fig 1B truly "rotation invariant" or related to cyclic permutations of the vector indices? It is unclear from the text. The figure does not depict a rotation (i.e. action by an orthogonal matrix), since the orbit is elliptical rather than circular and it is not centered at the origin.”** We follow the procedure and visualization from the original paper [3]. The reason the figure shows an elliptic orbit is because it is a projection of a circle from a higher space for 2d, therefore, it does not have to stay a zero-mean circle. The idea is that we have rotation equivariant vectors from the model, which consist of n blocks, where n is the number of rotations. Then these vectors are transformed to be rotation-invariant with the help of the cyclic permutations along rotations.
>
> We hope these comments shed some light and clarified our motivation, technical and design choices.
>
> [1] Lause et al. (2024) doi: 10.1101/2024.03.26.586728
> [2] Van Der Maaten (2009) AI-STATS, PMLR 5:384-391, 2009
> [3] Ustyuzhaninov et al., ICLR 2020
> [4] Klindt et al. (2017) doi: 10.48550/arXiv.1711.02653

---

> > ### Comment · Reviewer_mU1L · 2024-08-09
> >
> > Thanks for your response. I hope that you will include the references and clarifications that you've provided in the final form of the paper.

---

### Official Review · Reviewer_RvHn · 2024-07-18

**Soundness:** 3
**Presentation:** 3
**Contribution:** 3
**Rating:** 7
**Confidence:** 4

**Summary:**

The paper "Reproducibility of predictive networks for mouse visual cortex," explores the reproducibility of neuronal embeddings in the mouse visual cortex using deep predictive models. By introducing adaptive regularization and iterative feature pruning, the authors address key issues related to model overparameterization and provide a robust framework for achieving consistent functional representations of neurons. The work lays the groundwork for future research aimed at developing reliable and interpretable models for understanding neuronal function.

The primary goal is to investigate the stability and reproducibility of neuronal function embeddings derived from deep predictive models. These models aim to predict neuronal responses to sensory inputs and have been proposed to define functional cell types via unsupervised clustering. The paper addresses the concern that deep models are often highly overparameterized, leading to multiple solutions that can represent the same neuronal function, thereby questioning the reliability of embeddings for downstream analysis.

The paper demonstrates that L1 regularization, which was used in early models, is crucial for obtaining structured and consistent neuronal embeddings when newer readout mechanisms are used. A novel adaptive regularization scheme is introduced, which adjusts the strength of regularization for each neuron. This method improves the consistency of neuronal embeddings across different model fits while maintaining predictive performance. The paper proposes an iterative feature pruning strategy to reduce the dimensionality of performance-optimized models by half without losing predictive performance. This pruning improves the consistency of neuronal embeddings concerning clustering neurons.  The consistency of neuronal embeddings is evaluated using the Adjusted Rand Index (ARI) of clustering partitions across models, correlations of predicted responses across models, and the consistency of tuning indexes describing known nonlinear functional properties of visual neurons.

**Strengths:**

- Adaptive Regularization Scheme: The introduction of an adaptive regularization method that adjusts the regularization strength per neuron is a significant novelty. This approach helps achieve better consistency in clustering neuronal embeddings without compromising on predictive accuracy.

- Iterative Feature Pruning: The feature pruning strategy to address overparameterization in deep models is another novel contribution. By systematically reducing the model’s dimensionality, the authors enhance the robustness and consistency of the neuronal embeddings.

- Comprehensive Consistency Evaluation: The paper provides a comprehensive evaluation of model consistency across different dimensions (embedding clustering, predicted responses, and tuning curves). This thorough approach highlights the robustness and reproducibility of the proposed methods.

- The paper is well-written with generally sufficient referencing to the previous methods.

**Weaknesses:**

- Bias from Regularization and Pruning: While the adaptive regularization and pruning strategies improve consistency, they may introduce biases that affect the biological validity of the neuronal embeddings. Over-regularization, for instance, can reduce the model’s expressive power and lead to less biologically plausible representations.

- The study focuses primarily on models with a rotation-equivariant convolutional neural network (CNN) core. The authors acknowledge that other core architectures, such as regular CNNs or Vision Transformers, were not evaluated. This limitation means the findings may not generalize across different model architectures.

- Despite the improvements in clustering consistency, there is a trade-off between consistency and predictive performance. The pruning and regularization strategies, while improving consistency, sometimes result in a drop in predictive performance, which is not ideal.

- The introduction of adaptive regularization adds another layer of hyperparameters (e.g., the log-normal hyperprior parameter) that need to be carefully tuned. This increases the complexity of the model training process and may require significant computational resources.

- The focus on achieving high consistency in clustering and embeddings may overshadow other important factors, such as the interpretability and biological relevance of the model outputs. Balancing consistency with these factors is crucial for developing useful predictive models in neuroscience.

**Questions:**

- Is ARI a metric you developed or did it exist before? If it existed before, it requires referencing.

- Have you considered evaluating other core architectures, such as Vision Transformers or regular CNNs, to see if the findings generalize across different model types? What specific reasons led you to choose a rotation-equivariant CNN core for this study?

- Could you elaborate on the potential biases introduced by the adaptive regularization and pruning strategies? How do you mitigate these biases? Have you explored other regularization techniques or pruning strategies that might offer a better balance between consistency and predictive performance?

- Are there specific aspects of model performance that are most critical for maintaining biological plausibility?

- The adaptive regularization introduces additional hyperparameters. How do you approach hyperparameter tuning to ensure optimal model performance? What are the computational costs associated with this tuning process?

- Why did you choose the Adjusted Rand Index (ARI) and other metrics for evaluating consistency? Are there alternative metrics that might provide additional insights? How do you ensure that the chosen metrics accurately reflect the biological significance of the neuronal embeddings?

---

> ### Author Rebuttal · Authors · 2024-08-07
>
> Thank you for thoroughly reviewing our paper and providing valuable feedback. We are happy that you found the adaptive regularization scheme “a significant novelty”, iterative pruning as “another novel contribution” and our consistent evaluation procedure to “highlight the robustness and reproducibility of the proposed methods”.
> We have addressed all your comments and will incorporate them into our manuscript to strengthen it further:
> * **Is ARI a metric you developed?**
> ARI is a standard metric introduced by Hubert and Arabie [1] to measure how consistent two clusterings are. We will explicitly point this out and add the references.
> * **Have you considered evaluating other core architectures?**
> The motivation behind the rotation-equivariant core is that V1 neurons are orientation selective and neurons with all preferred orientations exist. Thus, any feature is likely to be extracted in multiple rotated versions [2,3]. We do not see a reason why our results should be specific to the rotation-equivariant core, though. To justify this intuition, we performed a few control experiments with a non-rotation-equivariant CNN core (Fig. 2 + 5 in attached PDF): The general pattern of results is the same: the adaptive regularization improves consistency of clustering and so does pruning feature maps. We hope these controls solidify your trust in our results.
> * **Could you elaborate on the potential biases introduced by the adaptive regularization and pruning strategies? How do you mitigate these biases? Have you explored other regularization techniques or pruning strategies that might offer a better balance between consistency and predictive performance?**
> Thanks for pointing that out. The tradeoff between cluster consistency and predictive performance was exactly the starting point of our paper, and the adaptive regularizer is a first answer that mitigates it by performing good along both axes. Of course every model has (implicit or explicit) biases. To assess these biases, we performed the in-silico experiments shown in Fig. 5 and Table 1. This analysis shows that regularized Gaussian readout models strongly shrink the responses and do not faithfully predict neuronal tuning w.r.t. orientation and other nonlinear phenomena, whereas factorized readouts produce more biologically plausible predictions. We hypothesized that this might be related to the factorized readout being able to adapt its regularization strength per neuron, which motivated the new adaptive readout. As the in-silico analyses show, this readout indeed produces biologically plausible tuning curves and does not sacrifice the predictive performance, while maintaining consistency.\
> Regarding your question about alternative regularization techniques: Yes we explored several other ways of adapting the strength per neuron by using L1 or L2 penalties instead of the lognormal prior, but these did not work well as many of the adaptive coefficients collapsed to zero, weakening regularization.\
> You mention in the weaknesses that *“The focus on achieving high consistency [...] may overshadow other important factors, such as the interpretability and biological relevance of the model outputs.”* We believe it is a misunderstanding, as biological relevance is also a key goal for us, as discussed above. Figure 5 and Table 1 address exactly this issue and suggest that the adaptive regularization does not only improve consistency, but also maintains biological relevance of the models.
>
> * **Are there specific aspects of model performance that are most critical for maintaining biological plausibility?**
> Rigorous, biologically meaningful evaluation remains an open question, but the presence of non-linear phenomena (tuning curves) is crucial. While current methods focus on explained variance/correlation, models with similar scores can differ in explainability [4]. To our knowledge, we conducted state-of-the-art evaluations for biological meaning by testing the tuning curves.
> * **The adaptive regularization introduces additional hyperparameters. How do you approach hyperparameter tuning […]? What are the computational costs […]?**
> In fact it introduces only one additional hyperparameter, because the overall strength $\gamma$ is also a parameter of the non-adaptive regularizer. The only one that needed to be optimized was $\sigma$, which we chose by a simple line search using five different values. The goal here was to keep the distribution of coefficients mean centered around 1 and clearly non-zero but not too restricted. Thus, the additional computational cost is manageable.
>
> * **Why ARI? Are there alternatives? Do they reflect the biological significance**
> ARI is the most popular metric for pairwise cluster comparisons (Section 2 in [5]), which we think is more appropriate than set-based methods. To ensure we are not missing anything, we also considered alternative metrics such as Fowlkes-Mallows, completeness, homogeneity, and v-measure (a particular case of mutual information) and repeated the analysis. The results show the same ordering of methods (e.g. Fig. 4 in attached PDF). \
> *Do the metrics reflect biological significance?* We reasoned that if neuronal embeddings determine the response function of a neuron in the model, then two neurons that have similar embeddings under one instance of a model, should also have similar embeddings under another instance of this model. Therefore, measuring similarity of clustering is a reasonable proxy.
>
> We hope these comments shed some light on our motivation, technical and design choiced. We also hope that it would help the reviewer to reconsider some criticism.
>
> [1] Hubert and Arabie (1985) doi: 10.1007/BF01908075
> [2] Ustyuzhaninov et al. (2022) doi: 10.1101/2022.02.10.479884
> [3] Ecker et al. (2019) doi: 10.48550/arXiv.1809.10504
> [4] Burg et al. (2021) doi: 10.1371/journal.pcbi.1009028
> [5] Vinh et al. (2010) doi: 10.1145/1553374.1553511

---

> ### Comment · Area_Chair_EnMd · 2024-08-12
> **Gentle reminder to respond to rebuttal**
>
> Dear Reviewer RvHn,
>
> As the author-reviewer discussion period is about to close, I would like to know your thoughts on the author rebuttal.  Especially since the other reviewers all currently leans towards acceptance, it would be extremely informative for me to know if you still maintain your original score following the rebuttal.  I would very much appreciate if you could reply to the authors before the close of the discussion (Nov 13 11:59 pm AoE).
>
> Gratefully,
>
> AC

---

### Author Rebuttal · Authors · 2024-08-07

We thank all reviewers for their constructive and positive feedback on our paper, highlighting its novelty and importance.

One concern was that our analysis might not generalize beyond our architecture choice of a rotation-equivariant neural predictive model. We now have performed additional experiments showing our results generalize to a more generic architecture as well (Fig 2 and 5 in attached pdf).

Another question was whether our analyses are biologically plausible. We now have thoroughly discussed in our answers to the reviewers that we verify biological validity through a number of experiments (Fig 5 and Table 1 in the original manuscript) and will revise our final paper to describe this more clearly.

We want to thank all reviewers once again for these and their other great comments - comments we were keen to incorporate as they further improve our paper.

---

### Decision · Program_Chairs · 2024-09-25

**Decision:**

Accept (spotlight)

**Comment:**

Reviewers were impressed by the paper's contributions in the problem of DNN modeling of biological neurons: the novel methods of iterative feature pruning and adaptive regularization, and well-designed and clearly motivated experiments, as well as the focus on addressing the reproducibility of learned embeddings.  Other strengths of the paper included testing of the biological plausibility of the tuning curves which enhanced the evaluation of the method in addition to more standard ML metrics.  The results on the non-rotationally invariant CNN added during the rebuttal further contribute to the potential generalizability of the results of the paper, which were originally shown only for rotationally invariant networks (due to biological considerations).

Some reviewer comments indicate that the paper could potentially benefit by adding more explanations and clarifying the motivation and technical details, which the authors have agreed to attempt in their revision.

This work has demonstrated a high level of technical sophistication and quality, and is likely to have high impact on of neuronal data modeling.  Given the size of the audience that would find this work to be of interest, I am recommending it for a spotlight presentation.